# Microfluidic Device Using Mouse Small Intestinal Tissue for the Observation of Fluidic Behavior in the Lumen

**DOI:** 10.3390/mi12060692

**Published:** 2021-06-13

**Authors:** Satoru Kuriu, Naoyuki Yamamoto, Tadashi Ishida

**Affiliations:** 1Department of Mechanical Engineering, School of Engineering, Tokyo Institute of Technology, Kanagawa 226-8503, Japan; 2Department of Life Science and Technology, School of Life Science and Technology, Tokyo Institute of Technology, Kanagawa 226-8503, Japan; n-yamamoto@bio.titech.ac.jp

**Keywords:** microfluidic device, small intestine, ex vivo, histology, embedded resin, sectioning

## Abstract

The small intestine has the majority of a host’s immune cells, and it controls immune responses. Immune responses are induced by a gut bacteria sampling process in the small intestine. The mechanism of immune responses in the small intestine is studied by genomic or histological techniques after in vivo experiments. While the distribution of gut bacteria, which can be decided by the fluid flow field in the small intestinal tract, is important for immune responses, the fluid flow field has not been studied due to limits in experimental methods. Here, we propose a microfluidic device with chemically fixed small intestinal tissue as a channel. A fluid flow field in the small intestinal tract with villi was observed and analyzed by particle image velocimetry. After the experiment, the distribution of microparticles on the small intestinal tissue was histologically analyzed. The result suggests that the fluid flow field supports the settlement of microparticles on the villi.

## 1. Introduction

The mammalian gastrointestinal (GI) tract harbors various types and tremendous amounts of gut bacteria (GB) [1,2,3]. Different types of GB distribute in the GI tract [4,5] and maintain their host’s health condition [6,7,8]. Among GI tissues, the small intestine (SI) controls immune responses, as over 60% of the host’s immune cells inhabit it. Immune responses are induced by a GB sampling process, the transportation of GB to GI tissue by fluid flow and responses with microfold cells (M cells) [6] or dendrites being extended by dendritic cells (DCs) [9,10]. In addition to the gut-associated lymphoid tissues (e.g., Peyer’s patch, which expresses M cells surrounded by SI’s unique finger-like structures, called villi [11]) [6], villi also have M cells and induce immune responses through a GB sampling process [12]. The mechanisms of the gut immune system were obtained from dissected specimens after an in vivo experiment that mainly used experimental mice [13,14,15]. This is because in situ visualization of interactions of the GB and the M cells/dendrites being extended by DCs in the SI tracts are difficult in in vivo studies. However, GB behaviors in the SI tracts are important for immune responses because they are triggered by the encounters of GB and M cells/DCs. In the SI tracts, GB distribution is decided by fluid flow, which controls the encounters of the GB and the M cells/DCs. Therefore, fluid flow fields inside the SI tracts are important. Therefore, technologies enabling the observation of the fluid flow field in the SI tract are required at the microscopic scale.

Microfluidic technologies have high potential for the direct and microscopic observation of the fluid flow field in luminal circumstances. This is because they can reconstruct biological systems on chips and build elaborate fluidic systems with biological samples, such as cells and tissues, under microscopic observation. An in vitro biomimetic “gut-on-a-chip” was proposed to mimic intestinal biological functions and enable microscopic observations [16,17,18,19,20,21,22]. Those chips commonly have two microchannels in parallel, separated by a flexible and a porous membrane. On both sides of the membrane, single or different kinds of cell (e.g., human colorectal adenocarcinoma cell (Caco-2 cell) and human umbilical vein endothelial cell (HUVEC)) are cultured over the long term (>7 days). To apply shear stress on the epithelial cells, continuous flow is perfused in both microchannels. This culture method mirrors intestinal biological features. With the gut-on-a-chip as a simulation of disease, drug tests were conducted [18,21]. However, the configurations were different in size and structure from the surface of the SI tract, which were assumed to be important for the fluid flow field in the SI tracts. On the other hand, microfluidic devices utilizing the natural features of full thickness intestinal tissues ex vivo were proposed [23,24,25,26]. Those devices have a sheet of a dissected intestinal tissue in the channel. They enable tissue culture for 3 days to monitor some functions in living intestinal tissues. In addition, microfluidic devices enable the observation of the intestinal tissue ex vivo [26]. However, the observations are only at the macro scale, and the analyses are performed only by the measurement of effluent from the outlet of the device. Although microfluidic technologies can study biological aspects of the gut, they do not focus on the studies of fluid flow in the gut at a microscopic scale.

For the observations of fluid flow field in the SI tract, especially around the villi at the microscopic level, a microfluidic device with the geometry of the SI tract is necessary. Here, we propose a microfluidic intestinal channel device (MIC) to reproduce and visualize the flow field inside an SI tract. MIC has a chemically fixed SI tissue dissected from a mouse. The SI tissue is used as microfluidic channel to visualize the flow field around its villi, although it is not an in vivo model but an in vitro one. In the fabrication and assembly of MIC, the handling of the SI tissue is the critical issue due to its small, soft and crooked characteristics. Modules to hold or to set the SI tissue were designed, and stable handling was achieved. With this device, we demonstrated the observation of fluid flow field around the villi at the microscopic scale by particle image velocimetry (PIV) and histological analysis of SI tissue.

## 2. Materials and Methods

### 2.1. Material and Reagents

Materials for device fabrication and assembly were obtained from the following suppliers: polydimethylsiloxane (PDMS) (Silpot 184 W/C, Dow Corning Toray, Tokyo, Japan), epoxy-based embedding resin (EPOX) (Technovit^®^ EPOX, Kulzer, Hanau, Germany), agar (Bacto™ Agar, Wako, Osaka, Japan) and liquid bandage (Mentholatum, Rohto, Osaka, Japan).

Reagents for experiments were obtained from the following suppliers: 4% paraformaldehyde (4% PFU) (Nacalai Tesque, Inc., Kyoto, Japan), fluorescent beads (Fluorescent Polystyrene Microspheres, 1.00 μm, Dragon Green, Bangs Laboratories, Inc., Fishers, IN, USA), anti-integrin antibodies conjugated to AF555 (Anti-integrin αvβ5 Antibody, Alexa Fluor^®^ 555 Conjugated, Bioss, Woburn, MA, USA).

### 2.2. Tissue Preparation

Balb/c mice (female) aged 9 weeks were obtained from Charles River Laboratory, Japan (Yokohama, Japan), and housed in the animal facilities at Tokyo Institute of Technology. Animal experiments were approved by the Animal Experiment Committees of Tokyo Institute of Technology (authorization numbers D2019006-2), and carried out in accordance with relevant guidelines. Full thickness SI tissues taken from a part between duodenum and jejunum were dissected from mice, and immediately fixed in 4% *w/v* paraformaldehyde solution. The fixation prevented the degradation caused by digestive enzymes (e.g., trypsin, chymotrypsin, and many amino-/carboxy-peptidases) released from the SI tissues themselves at 37 °C, which is a suitable temperature for GB. The fixation preserved the geometry of the villi, which was important for the flow field inside the SI. While it might change the surface condition of charges, the flow field was not affected at a micro scale. Although the density of GB and immunologic activity are different for each part of SI tissue (e.g., duodenum, jejunum and ileum), they are mainly covered by villi and have similar surface geometries of villi.

### 2.3. Image Acquisition and Processing

Bright-field and confocal fluorescent images were captured by an inverted microscope (IX83, Olympus, Tokyo, Japan) equipped with a high-speed scanner (CSU10, Yokogawa, Tokyo, Japan) and an electron multiplying charge coupled device (EMCCD) camera (iXon Ultra, Andor, Tokyo, Japan). For the excitation light source used for confocal microscopy, solid-state lasers (wavelength: 488 and 561 nm, TAC, Saitama, Japan) were used. The confocal fluorescence images were saved in 16-bit TIFF format using a capture software (iQ3, Andor, Tokyo, Japan). The images were processed and quantified by Image J (National Institute of Health, Bronx, NY, USA).

### 2.4. PIV

Suspension of tracer particles (fluorescent microbeads of 1.0 μm in diameter) were perfused in a microchannel. The tracer particles in flow were captured at an interval of 1 s. The images were captured in 1280 pixels × 720 pixels. Two serial images were compared by dividing them into interrogation areas of 64 pixels × 64 pixels. The interrogation areas were cross-correlated with each other, pixel by pixel, using a function of Image J, resulting in instantaneous velocity fields.

### 2.5. MIC

#### 2.5.1. Design of the MIC

A schematic illustration of the MIC is shown in Figure 1. The MIC consisted of three modules: container, connector and channel. The container module worked as the guide of the SI tissue. A SI tissue was placed on the guide in the container module, whose tube-like structure worked as a microfluidic channel (SI channel). The container module, including the SI tissue, could be sectioned after experiments. The container module also held the position of the connector module. The connector module worked as an inlet and an outlet of the SI channel. Both ends of the SI tissue were inserted into the connector module for liquid perfusion and glued to prevent liquid from the leakage. The channel module was the bottom part of the MIC. The channel module worked as the substitute of the sliced part of the SI tissue for observations inside. To keep the transparency for observation, all the modules were made of PDMS or EPOX.

To achieve the MIC, the container module, connector module and channel module were separately designed. The typical dimensions of mouse SI tissue were 10 mm in length and 3 mm in diameter in our experiments. The guide for the SI tissue was 3.5 mm in width, which was slightly larger than typical diameter of the SI tissues. The height of the guide was 1.5 mm, half of the diameter, because the half of the SI tissue was removed to make an opening for the observation of the inner structure of the SI channel. Furthermore, histological analysis was necessary for the SI tissue after experiments. The SI tissue needed to be sliced inside the container module to suppress any changes and damages in the sectioning process. For the sectioning of the SI tissue set on the guide, the container module was made of EPOX, which is a suitable material for sectioning and microfluidic devices. Mechanical strength of the guide was weak due to the thin structures, especially the ceiling thickness of 0.1 mm (Figure 2A). This is because the guide for the SI tissue needed to be isolated from the container module and transferred to the sectioning process. To reinforce the thin structures, PDMS was cured on the thin structure considering its transparency and easy removal from EPOX.

To use the SI tissue as a microfluidic channel, inlet and outlet were the key modules for the stable liquid perfusion. The connection with a tight seal between the SI tissue and accessories (i.e., tubes or stainless pipes) was important for the stable liquid perfusion. Therefore, we designed the connector module which tightly hold both ends of the SI tissue (Figure 2B,C). The inlet connector had a double pipe (i.e., coaxial stainless pipes of 1 mm and 3 mm in diameter) (Figure 3). The stainless pipe was 1 mm in diameter and had a spacer of 2 mm in diameter at the tip. The pipe that was 1 mm in diameter was inserted into the SI tissue, and the SI tissue was covered by the pipe 3 mm in diameter. The outlet connector only had a single stainless pipe of 3 mm in diameter. This is because the stainless pipe of 3 mm in diameter could reduce fluid resistance, leading to good drainage (i.e., the fluid resistance in the case of the pipe that was 3 mm in diameter was 27 times lower than that of the inlet pipe that was 1 mm in diameter). The pipes and SI tissue were tightly sealed by a liquid bandage.

The channel module was the bottom part of the MIC. The dimensions of the channel should be close to the sliced SI tissue and designed to be a rectangular cross section of 10 mm (length) × 3 mm (width) × 500 µm (height). This is because, for the microscopic observation of the inner structures of the SI channel, the bottom of the channel module should be flat. Further, the channel module should be transparent. Although PDMS was a good choice from the point of transparency, the conventional permanent bonding of PDMS microfluidic devices was not applicable for sealing. This is because the SI tissue is weak against dryness or heat. Therefore, the components of MIC should be mechanically assembled by sealing with screws. For the tight mechanical sealing, PDMS was a good material from the point of softness and adhesion to EPOX structures [27].

#### 2.5.2. Fabrication Procedure of the MIC

The fabrication procedures of the container module and connector module are described in Figure 3. The container module and connector module were fabricated by conventional soft lithography. As for the fabrication process of the container module, EPOX (base EPOX to curing agent ratio was 2:1 by weight) was casted on molds and cured for 3 h at 100 °C in an air-vented oven (Figure 3A). The upper mold was detached (Figure 3B) and PDMS (base PDMS to curing agent ratio was 10:1 by weight) was casted on a thin structure of the container and cured for 3 h at 100 °C in an air-vented oven (Figure 3C). The container module was detached from the mold (Figure 3D). As for the fabrication process of the connector module, PDMS was casted on molds for an inlet and an outlet and cured for 3 h at 100 °C in an air-vented oven (Figure 3E). The cured PDMS were detached from the molds (Figure 3F). Stainless pipes were incorporated in each connector (Figure 3G). Finally, the container and connectors were assembled (Figure 3H). As for the fabrication of the aforementioned molds and holders, polyacetal plates were made by a milling machine. The molds were polished with abrasive compound (PiKAL NIHON MARYO-KOGYO) to remove residual traces of the milling. They were designed by three-dimensional computer aided design software.

#### 2.5.3. Assembly Procedure of the MIC

The assembly procedure is described in Figure 4. A SI tissue was set on the guide (Figure 4A). The SI tissue and connectors were glued by liquid bandage. The SI tissue was inflated by air. The inflation by air made the SI tissue straight and smooth to avoid difficulties in the slicing process of the SI tissue due to its softness and crooked shape, such as wrinkles, waves and curves (Figure 4B and Appendix A). The inflated SI tissue was embedded with pre-gel solution (agar to deionized (DI) water 5% *w*/*v*). The solution was gelated within a few seconds at room temperature (Figure 4C). Here, the concentration of the pre-gel solution was determined by the performance of holding and slicing the SI tissue in the following step. The agar also prevented the SI tissue from drying. The excessive volume of the agar and SI tissue were sliced by a blade (Figure 4D). The agar embedding made the handle of the SI tissue easy. The container module and connecter module including the SI tissue were assembled by holders, and they were screwed (Figure 4E). Note that the outlet connector was plugged to avoid the influx of liquid bandage and pre-gel solution into the SI tissue in the processes of tissue setting and agar embedding.

#### 2.5.4. Experimental Procedure Using MIC

The suspension of fluorescent microbeads (1.0 × 10^7^ particle/mL) was manually introduced into the SI tissue with a syringe. The MIC was placed in the manner of Figure 5A for 1.5 h for the settlement of the microbeads on the surface of the villi. DI water was introduced into the SI channel at a flow rate of 7 µL/min (Figure 5B). The flow rate was sufficient for the wash-out of the microbeads, which was calculated by flow rate and cross-sectional area [28]. Ethanol in a 95% concentration was perfused in the SI channel for 1 h at a flow rate of 5 µL/min for dehydration (Figure 5C). A hole of 3 mm in diameter was punched with a biopsy punch on the channel module. EPOX was introduced into the SI channel via the hole. The SI channel was embedded by EPOX and cured at room temperature overnight (Figure 5D). The channel module and PDMS reinforcement of the container module were removed from the MIC, and the guide was manually taken apart from the container module (Figure 5E). The EPOX-embedded SI tissue was sectioned by a thickness of 30 μm for histological observations (Figure 5F).

## 3. Results and Discussion

### 3.1. Fabricated MIC

The fabricated container module, connector module and MIC are shown in Figure 6A–C, respectively. The villi of the SI tissue in MIC were visualized through an opening of the SI tissue and the channel module with a microscope. The villi were stained with red-fluorescent labeled anti-integrin antibody, and visualized at different heights (i.e., bottom, middle and top. Figure 6D). The structures of the villi were well maintained in the MIC.

### 3.2. Analysis of the Microbeads Distribution in SI Channel

The settled fluorescent microbeads on the surface of the villi were observed after the wash-out by the flow of DI water (Figure 7, Appendix A). The distributions of microbeads (Figure 7A) and villi (Figure 7B) were observed. We found that the microbeads did not cover the villi themselves but the base of the villi (Figure 7C).

### 3.3. Analysis of Flow Field in SI Channel

To analyze the flow field in the SI channel under the continuous flow, DI water was perfused into the SI channel at a constant flow rate as a simple model after the settlement of microbeads. The size of the microbeads was comparable to that of GB. The motions of the microbeads inside the SI channel were observed at the bottom, middle and top (Figure 8). PIV analysis was conducted against the images. The velocity fields in the planes at the bottom and middle were fast (26.3–48.8 μm/s) and stable (Figure 8A,B), while the velocity field of the top plane was slow (1.3–6.3 μm/s) (Figure 8C). From the defecation time in mouse, the flow speed of GB was empirically estimated around 2.8–14 μm/s. The speed of the microbeads flowing in the SI channel was comparable to that in the GI tract.

The flow velocities at different planes follow the basic trend of three-dimensional velocity field of the flow in a pipe; that is, the velocity was maximal at the center of the pipe and decreased near the pipe wall. Furthermore, the microbeads flew around the villi at the top, which was different from the flows at the bottom and middle. The flow between villi was slow but some of them headed for villi, while the velocity over villi was ~1 μm/s. This result suggests that flow around the villi may transport and settle GB on the surface of the villi.

### 3.4. Transition of the Fluorescent Beads Distribution around the Villi against Time

We evaluated the temporal transition of the fluorescent intensity in the SI channel. The change in the fluorescent intensity was caused by the density change in the fluorescent microbeads around the villi, namely at the top in Figure 8C (Figure 9). Fluorescent intensity was measured every 5 min. Fluorescent intensity gradually decreased to 71 and 52% in 15 and 30 min, respectively (Figure 9B,C), compared to that in the initial state (Figure 9A). However, the fluorescent beads should not move a great deal, at least according to the PIV analysis in Figure 8C. The fluorescent microbeads on the surface of the villi could gradually fall as a result of gravity rather than the flow.

### 3.5. Histological Analysis of the Microbeads Distribution

The MIC was disassembled and the guide, including the SI tissue, was manually isolated (Figure 10A), as explained in Section of 2.5.1. The guide channel was sectioned at a thickness of 30 μm, and a series of slices was acquired (Figure 10B). The SI tissue in the MIC was histologically analyzed after the experiments. The observation of the series of slices revealed that the fluorescent microbeads were attached on the side surface of the villi, and they were especially trapped in the pit-like structures (Figure 11). This result suggests that the geometrical features of the villi may affect the trapping performance of microparticles for the settlement of GB.

## 4. Conclusions

We developed a microfluidic device with mouse SI tissue as a microfluidic channel. The microfluidic device consisted of a container, connectors and channel modules, and it achieved easy handling in the mouse SI tissue. With this device, the villi of the SI tissue and fluorescent microbeads were observed clearly under liquid flow at the microscopic level. We perfused the steady flow in the SI channel and analyzed the flow velocity field inside the SI tissue by PIV analysis. Some of the microbeads headed to the villi and attached to the side surface of the villi. This result suggested that the flow field may play an important role in the settlement of GB on the villi.

For future work, we will perfuse mucus-like liquid [29,30] or isotonic saline suspending living GB to understand the roles of the flow field in forming GB distribution in the SI tissue using this microfluidic device, and we will apply an inconstant flow as a simulation of peristatic motion. Furthermore, although the SI tissue used in our study was fixed and lost its biological activities, the superficial molecules on mucosal tissues and the shape of villi were properly maintained. We plan to study the intermolecular affinity between the intestinal microbial community and different parts of the SI tissue (e.g., duodenum, jejunum and ileum) with an MIC, as the different geometry of the villi may affect the fluid’s behavior.

## Figures and Tables

**Figure 1 micromachines-12-00692-f001:**
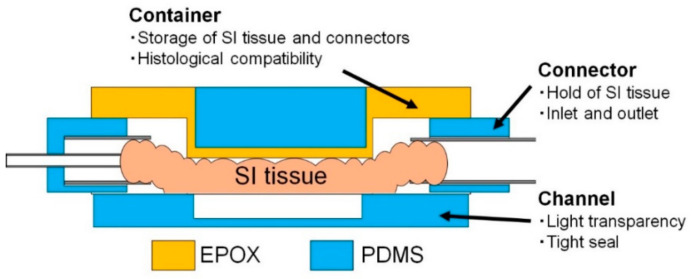
Schematic illustration of the MIC. The container module stores the SI tissue with an opening for microscopic observation. The connector module holds both edges of the SI tissue. The channel module is the bottom part of the MIC, a substitute of the SI tissue.

**Figure 2 micromachines-12-00692-f002:**
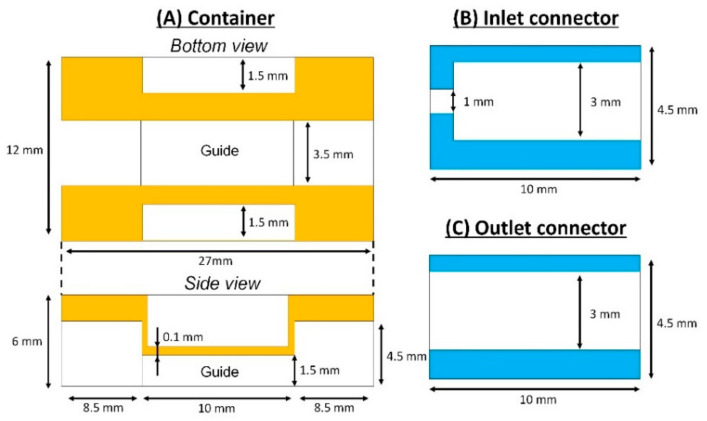
Schematic illustrations of the container and connectors with dimensions: (**A**) Container; bottom view (upper) and side view (lower). (**B**) Inlet connector. (**C**) Outlet connector.

**Figure 3 micromachines-12-00692-f003:**
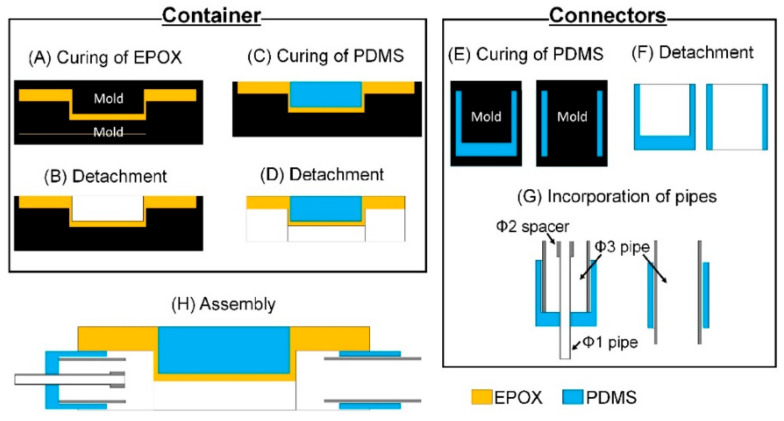
Fabrication procedure of the container and connectors. Fabrication procedure of the container: (**A**) Curing of EPOX. (**B**) Detachment. Upper mold is detached. (**C**) Curing of PDMS. PDMS is filled in the hollow of the cured EPOX and cured. (**D**) Detachment. Lower mold is detached. Fabrication procedure of the connectors. (**E**) Curing of PDMS. (**F**) Detachment. (**G**) Incorporation of pipes. Stainless pipes (diameter: 1 or 3 mm) are incorporated. (**H**) Assembly.

**Figure 4 micromachines-12-00692-f004:**
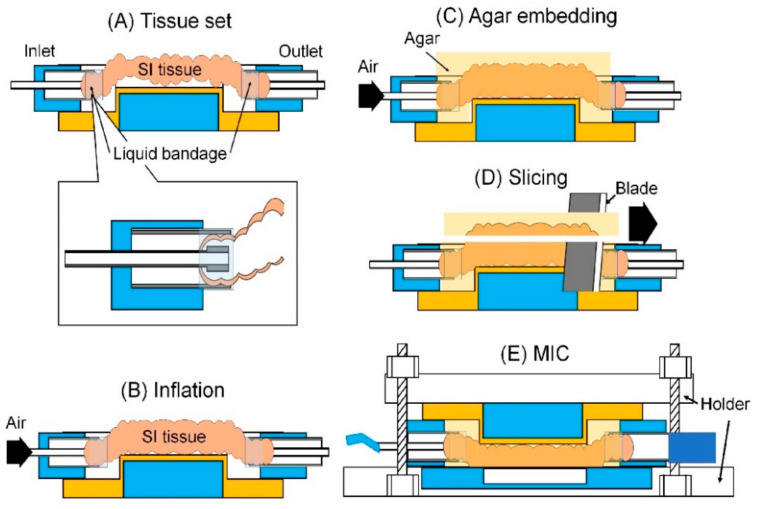
Assembly procedure of the MIC: (**A**) Tissue set. (**B**) Inflation. Air is pumped into the SI tissue. (**C**) Agar embedding. (**D**) Slicing. Excessive volume of the agar including SI tissue is sliced. (**E**) MIC.

**Figure 5 micromachines-12-00692-f005:**
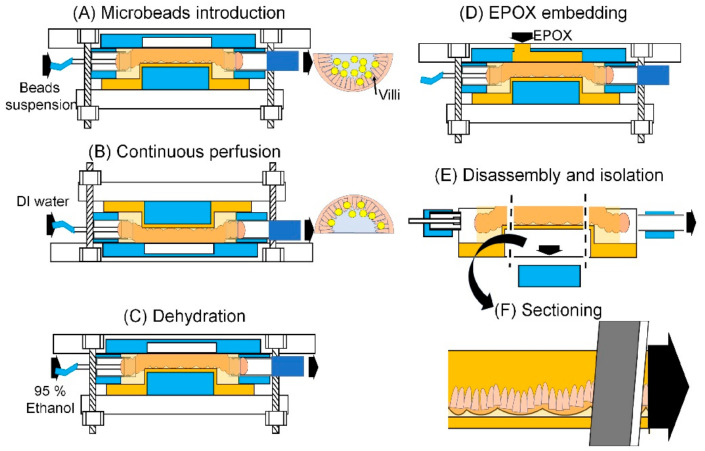
Experimental procedure using the MIC. (**A**) Microbeads introduction. (**B**) Continuous perfusion. DI water is perfused in the SI channel at the flow rate of 7 µL/min. (**C**) Dehydration. Ethanol at a 95% concentration is perfused in the SI channel for 1 h at a flow of 5 µL/min. (**D**) EPOX embedding. (**E**) Disassembly and isolation. (**F**) Sectioning. The guide including embedded SI tissue is sectioned with a thickness of 30 µm.

**Figure 6 micromachines-12-00692-f006:**
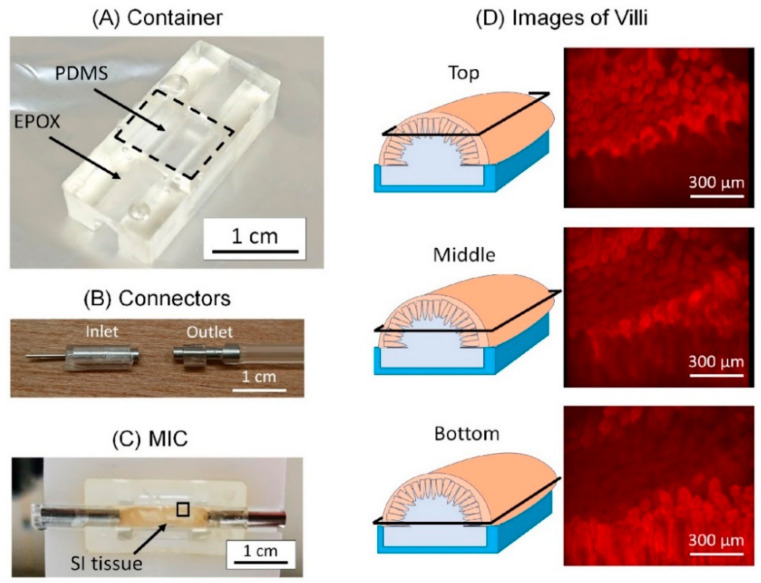
Fabricated modules and the MIC. (**A**) Fabricated container. Location of PDMS reinforcement is indicated by black dashed line. (**B**) Fabricated connectors for the inlet (left) and outlet (right). (**C**) MIC. (**D**) Images of villi in the black square in (**C**). The villi were captured at the top, middle and bottom. The villi were stained by red fluorescent-labeled anti-integrin antibodies.

**Figure 7 micromachines-12-00692-f007:**
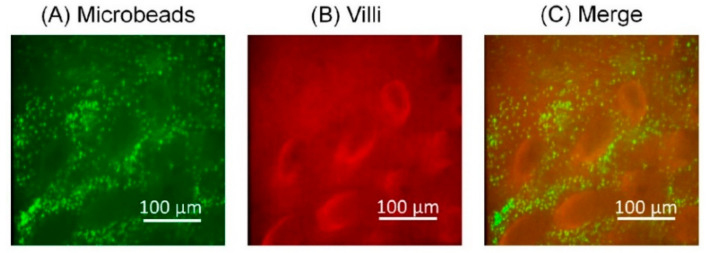
Observation of inner structures of the SI tissue in the MIC. (**A**) Microbeads. They are indicated by green fluorescent. (**B**) Villi. They are indicated by red fluorescent. (**C**) Merge.

**Figure 8 micromachines-12-00692-f008:**
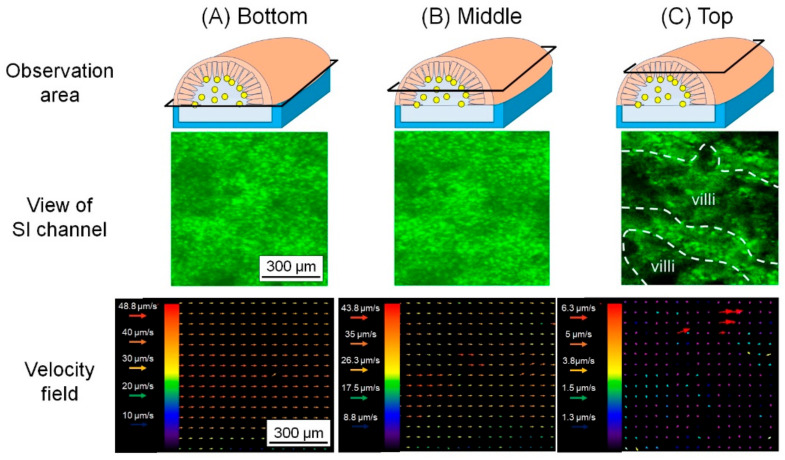
PIV analysis in the SI tissue. The fluorescent microbeads were tracked to calculate the velocity vector in the SI channel. Top column: the schematic of the observation area at the (**A**) bottom, (**B**) middle and (**C**) top. Middle column: view of the microbeads in the SI channel at the (**A**) bottom, (**B**) middle and (**C**) top. Lower column: the velocity fields at the (**A**) bottom, (**B**) middle and (**C**) top. In the image at the top, the villi are surrounded by white dash line.

**Figure 9 micromachines-12-00692-f009:**
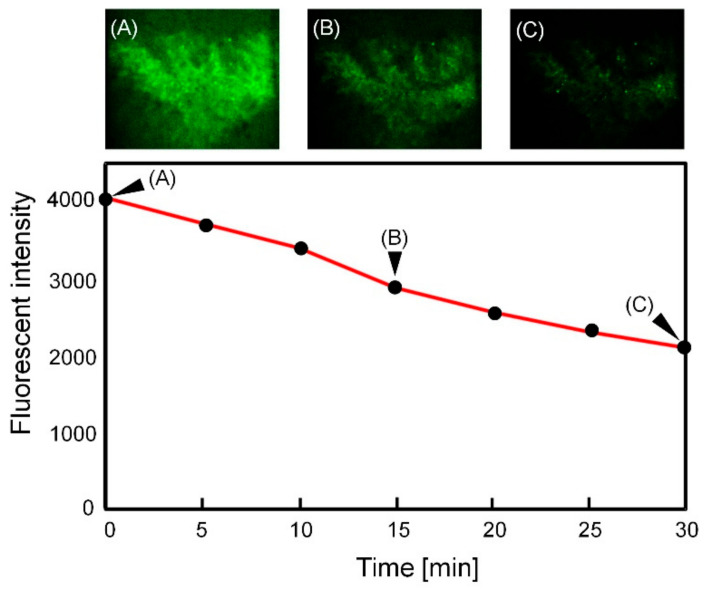
Transition of fluorescent intensity around the villi as a function of time. Fluorescent images were captured every 5 min: (**A**) Fluorescent image at 0 min, (**B**) 15 min and (**C**) 30 min. Black arrows in the graph indicate that the data acquired from fluorescent images (**A**), (**B**) and (**C**).

**Figure 10 micromachines-12-00692-f010:**
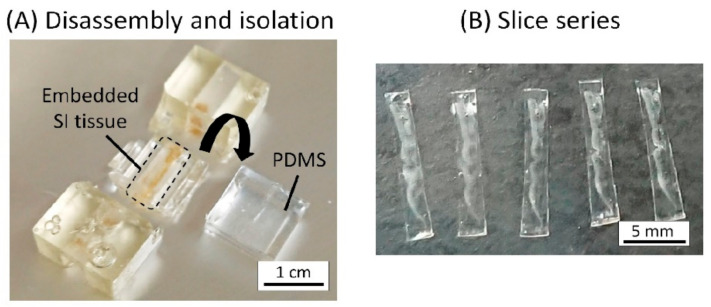
Sectioning of the guide channel including embedded SI tissue by EPOX: (**A**) Disassembly and isolation. The PDMS reinforcement was removed, and the SI tissue embedded in the container by EPOX was isolated. (**B**) Slice series. The embedded SI tissue was sectioned by the thickness of 30 µm.

**Figure 11 micromachines-12-00692-f011:**
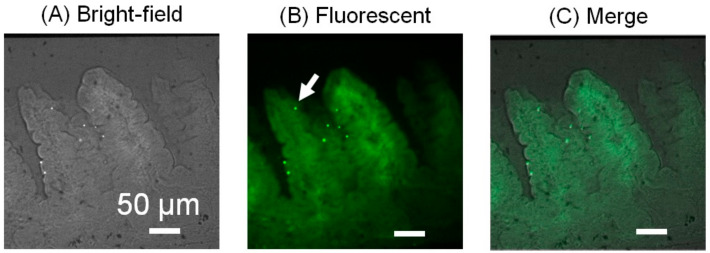
Histological analysis of the SI tissue: (**A**) Bright-field image, (**B**) fluorescent and (**C**) merged images of the cross-sections of the villi. The bright dots represent the fluorescent microbeads attached to the villi. One of the fluorescent microbeads is indicated by the white arrow in (**B**). Villi are indicated by autofluorescence.

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
