# Peer review of "Microfluidic Device Using Mouse Small Intestinal Tissue for the Observation of Fluidic Behavior in the Lumen"

_micromachines, 2021, doi:10.3390/mi12060692_

Round 1
Reviewer 1 Report
Review for Micromachines of Manuscript 1226738: “Ex vivo microfluidic device using mouse small intestinal tissue for the observation of the fluidic behavior in the lumen” by Satoru Kuriu and colleagues.
This manuscript reports on a fluidic device that holds paraformaldehyde fixed tissue and allows for the visualization of fluorescent microbeads in DI water. It is not clear what this preparation models.
There are several concerns outlined below:
- The 1st paragraph of the manuscript notes that the gut microbiome is a big component of a gut ecosystem. However, the design of the device and procedures used would eliminate this component by fixation, use of ambient air (toxic to anaerobic bacteria), and finally DI water as a hypo-osmotic solution would change the morphology of everything microbial or host - fixed or not.
- The article claims to observe “dynamic fluid behavior around the villi at the micro scale by particle image velocimetry (PIV) and the histological analysis of the SI tissue.” As noted in 1, the tissue preparation would preclude any dynamic behavior that was relatable to live intestinal tissue. Further, a key part of the ileum (only 1 component of small intestine) that is optimized for immunological surveillance are Peyer’s patches, which do not have villi and are not noted anywhere in the manuscript (except in titles of 2 papers cited in the references).
- The contrast and resolution in all of the figures of intestinal tissue are inadequate.
- There are no analyses to verify that any of the assessments provide useful information.
Author Response
To reviewer #1
We appreciate Reviewer #1 for taking the time to review our manuscript and give us his/her valuable comments. We have considered all the comments and have made appropriate changes to the manuscript.
Please see the attachment.
Sincerely.

Reviewer 2 Report
This report describes creation of a microfluidic chip using explant mouse intestine. The results show that microbeads flow with a different speed at the different parts of the tissue. The novelty of their research is that the tissue can remain fixed in the device and processed directly for sectioning and morphology analysis. Overall, the work is interesting but there are significant problems that need to be addressed.
Major comments:
- The introduction starts with introducing the microbiota and the gut immune system as major components of the hosts health. However, the gut microbiota or immune response were not mentioned in the results or conclusion. Also not all information is correct, e.g. dendritic cells are present in the lamina propria of the intestinal wall but not on the luminal surface. As the novelty of the manuscript is in the technique of using ex vivo tissue and being able to morphologically study it, the start of the introduction does not seem to fit the purpose of the article. Reading about the gut architecture would be an appropriate addition.
- Line 31: it is not clear what ‘dynamic interactions’ mean. Is this a mechanistical or biological interaction?
- In line 38 the authors use Gut-on-a-chip to refer to many different gut-on-a-chip devices, but the capital letter in Gut-on-a-chip suggest that this is a specific chip. Furthermore, try to incorporate the most original papers, and/or refer to reviews (e.g. of Bein et al., 2018 and Donkers et al., 2020) that accurately describe the recent developments in the field.
- In line 41 the authors describe that different type of cells were cultured on each side of the membrane. Can you specify which cell types? Please consider that there are also gut-chips that only culture one cell type.
- Line 48, please specify ‘long-term’. Certain cell-based chips can (co-)culture longer than tissue-based chips.
- In line 44 the authors refer to reference 20 (Suh et al., 2019) to state that the previously discussed gut-on-a-chip models ‘lack the natural configuration of villi’. However, the paper of Suh et al., writes about lacteal integrity and villus macrophages and not about the presence or absence of villi in gut-on-a-chip models. Furthermore, the paper cited in reference 19 (Kim et al., 2013) clearly states that villi-like structures are spontaneously formed in their gut-on-a-chip device, contradicting the statement made in the current manuscript.
- With respect to the gut-on-a-chip device developed by the Ingber group, please also cite their first reference from 2012 (Human gut-on-a-chip inhabited by microbial flora that experiences intestinal peristalsis-like motions and flow by Kim, Huh, Hamilton and Ingber).
- In the materials and methods it is described that the SI taken from the mice are immediately fixed in 4% formaldehyde, thereby the tissue is no longer alive. However, how can tissue function be studied ex vivo if the tissue is not alive? Especially when the aim is to study the interaction between the tissue and the microbiota? Can the same setup not be used for freshly isolated tissue that was not fixed with 4% formaldehyde? Is it possible to demonstrate this?
- Which part of the SI of the mice was taken for the studies? Was a specific region used, e.g. jejunum, and was this region the same between all different studies?
- How were the PIV analyses done? It seems not to be described in the Methods section.
- Line 118-119: Unclear why the channel should be made of EPOX for sectioning and not of any other material
- Unclear from figure 1 what is the guide channel? At some parts in section 2.4.1 there is ‘channel’ written and sometimes ‘guide channel’ are they the same or different?
- In section 2.4.3 it is described that the tissue is scraped. Although scraping does mean that something is removed, ‘scraped tissue’ was confusion as it sounds like the removal of the inner layer of cells and not cutting the tissue in half along the longitudinal axis as was done here. Therefore it might be useful to reconsider the word scraping.
- What is the aim of the microbead experiment?
- In line 235 the defecation speed in mouse intestine was shown with the statement that the speed of the microbeads was comparable. However, the speed of the microbeads is ~2-3 fold higher than the defecation speed in the bottom and middle of the tissue in the chip.
- Figure 8: Why are the villi in the middle left and middle middle picture not surrounded by a white line? It seems as if both this pictures show two or three spots with a lower intensity of green dots, can they be the villi?
- Could you please provide some information about the reproducibility of the technique, e.g. how many assemblies with SI tissue were successful, and are the results shown in Figures 6, 7, 8, 9, 11 n=1 experiments or a representative from multiple experiments?
- The conclusion summarizes the results but does not indicate the lessons learned and future perspectives. It would be nice to read what the authors think would be possible applications with their newly developed technique. In the conclusion the dynamic interaction with GM and the tissue are not mentioned, which were important in the introduction.
Minor comments:
- The English language is not always correct, especially in the introduction. It would be advised to let it read by a native speaker to correct for grammar errors in the sentences, e.g. plural/single form, use of articles, tense etc. Some examples are given below but more can be found.
- Line 21, 23, 236: ‘GI track’ should be ‘GI tract’
- Along the paper : ‘ex vivo‘, ‘in vitro’, ‘in vivo’, should be written in Italic
- Line 40: ‘had’ should be ‘have’, as this design of the chips still exist.
- Line 117-119: Sentences are written as ‘should be’ but the Methods section should be written in the past tense as it is a description of what has been done and not what should be done
- Line 128: Sentence should be written in past tense so ‘designed’ instead of ‘design’
- Line 210: ‘were shown’ should be ‘are shown’ as it describes something that is still present, namely the pictures are presented in Figure 6A-C
Author Response
To reviewer #2
We appreciate Reviewer #2 for taking the time to review our manuscript and give us his/her valuable comments. We have considered all the comments and have made appropriate changes to the manuscript.
Please see the attachment.
Sincerely.

Reviewer 3 Report
The manuscript presents a microfluidic device with pre-fixed mouse small intestine for investigation of flow field around the villi visualized by using fluorescent particles. The work is partly valuable for the development of organ-on-a-chip. Some important contents and manuscript revisions are required before consideration of possible publication.
- What did the authors mean about “Ex vivo microfluidic device”. Ex vivo intestinal tissue? Correct it.
- The cells in fixed mouse small intestine are all dead. So how did the results from this in vitro model mirror the condition in vivo. Why didn’t the authors perform a culture-based investigaiton.
- The analysis associated with microbead dynamics was insufficient. There was not a convincing summary for fluid investigation.
- The images in Figure 10 are unclear. Provide the new pictures with high resolution.
Author Response
To reviewer #3
We appreciate Reviewer #3 for taking the time to review our manuscript and give us his/her valuable comments. We have considered all the comments and have made appropriate changes to the manuscript.
Please see the attachment.
Sincerely.

Reviewer 4 Report
This work by Kuriu et al describes a microfluidic platform for studying post mortem intestinal fluid. the authors integrate the tissue and cut it open to enable observation, through microscopy. although interesting the authors don't clearly state the insight obtained by analyzing the fluorescent beads that are retained in the surface of the tissue. In general, I think the concept is interesting and can be used as a platform, nevertheless, the authors need to expand the discussion of potential future work and challenges. I would accept the manuscript after the authors improve upon those points.
Author Response
To reviewer #4
We appreciate Reviewer #4 for taking the time to review our manuscript and give us his/her valuable comments. We have considered all the comments and have made appropriate changes to the manuscript.
Please see the attachment.
Sincerely.

Round 2
Reviewer 1 Report
The response and the text now contains the phrase "dissected small intestine is immediately collapsed by the self-digestion". It is unclear what this means. All tubular structures in a body collapse when there is no fluid in them, however, that does not imply immediate "self-digestion".
There are still issues of what utility this model has. The response did not note the impact of DI water, which would cause swelling issues for the tissue (possibly evident in figure 6) and possibly for the beads. The fixation of the tissue would also significantly impact charge on the tissue surfaces and this is not addressed either.
The introduction talks about gut bacteria and immune system interactions and then the methods say that the tissue is taken from between duodenum and jejunum, which have significantly less immune components than the ileum. The introduction discusses a number of live models where relevance is questionable, and then suggests that fluid flow modeling is needed in fixed tissues. However, fluid flow in the intestinal lumen is NOT constant and is driven by contractions that don't happen after fixation.
Author Response
We really thank you for your valuable comments.
Response to Reviewers’ Comments
To Reviewer #1
We appreciate Reviewer #1 for taking the time to review our manuscript and give us his/her valuable comments. We have considered all the comments and have made appropriate changes to the manuscript. All the changes are highlighted in red. Our point-by-point response appears below, in which we first echo the Reviewer’s comments (shown in italic) and then respond to them:
Reviewer’s Comment:
- The response and the text now contains the phrase "dissected small intestine is immediately collapsed by the self-digestion". It is unclear what this means. All tubular structures in a body collapse when there is no fluid in them, however, that does not imply immediate "self-digestion".
Authors’ response:
As the reviewer pointed out, tubular living structures in a body collapse when the collected tissues are not in culture medium. In addition to the degradation, some of the gastrointestinal surface tissues secreted their digestive enzymes, and therefore, they are digested in a few minutes after the collection from mice at room temperature. Small intestinal tissue is a kind of the gastrointestinal tissues and has this degradation. This degradation was described as “self-digestion” in the previous revision. However, we understood that only the word “self-digestion” was not clear, and added the explanation of the degradation caused by the secreted digestive enzymes in the section 2.2.
Reviewer’s Comment:
- There are still issues of what utility this model has. The response did not note the impact of DI water, which would cause swelling issues for the tissue (possibly evident in figure 6) and possibly for the beads. The fixation of the tissue would also significantly impact charge on the tissue surfaces and this is not addressed either.
Authors’ response:
The model developed in this study used a fixed small intestinal tissue as a microfluidic channel. Its morphology, such as villi, was preserved because the small intestinal tissue was fixed immediately after the dissection. It had the function of reproducing and visualizing liquid flow field inside the small intestinal tract. We added the explanation in the introduction.
The fixation treatment performed in this study is used in usual histological studies in biology. It is widely believed that it prevents not only biochemical degradation of proteins but also geometrical changes of the small intestine. Therefore, we also believe that the swelling due to DI water is suppressed as much as possible. In addition, from the cross-sectional observation, the appearance seems similar to general sections of small intestinal tissue, and therefore, we believe that there is no expansion caused by the swelling of DI water. About microbeads, it is made of polystyrene, and the diameter kept constant. We added the explanation of the geometrical preservation of villi in the section 2.2.
As the pointed out, fixation can electrically affect the condition of the villus surface. However, the influence on the fluidic flow field around villi in our experiment caused by the change of the surface charge condition can be negligible, because the range of the influence is, at most, several nm from the surface considering the thickness of the electric double layer. We added the explanation in the section 2.2.
Reviewer’s Comment:
- The introduction talks about gut bacteria and immune system interactions and then the methods say that the tissue is taken from between duodenum and jejunum, which have significantly less immune components than the ileum. The introduction discusses a number of live models where relevance is questionable, and then suggests that fluid flow modeling is needed in fixed tissues. However, fluid flow in the intestinal lumen is NOT constant and is driven by contractions that don't happen after fixation.
Authors’ response:
This study examines the flow inside the small intestine, and therefore, geometry in the small intestine is important. As the reviewer pointed out, the ileum is an important organ in the gut immune system, including Peyer's patches. In the ileum of a mouse, there are only several Peyer’s patches, and most of the surface is covered with villi. The duodenum and jejunum also have an immune response, and most of them are also covered with villi. Therefore, in terms of geometry, the ileum, duodenum, and jejunum are similar. Based on the above consideration, there is no significant difference in flow fields inside the tracts among duodenum, jejunum, and ileum. We added the explanation in the section 2.2.
According to the reviewer’s comment, it was difficult to understand the relationship between the purpose of this research and the microfluidic device in the Live model in the introduction. For intestinal studies, interaction between an intestinal tract and bacteria is important. The interaction can be explained by flow of bacteria in the intestine and the response to bacteria attached to the intestine. To study the flow inside GI, microfluidic technology is important. In our survey, we could find microfluidic researches of GI. They are the live models for simulating the intestinal environment (gut-on-a-chip) and applying intestinal tissue (ex-vivo style device). The live models focused on the biological activity. However, we could not find research to study a flow in the small intestine, which is important for the flow of bacteria in the intestine. Some of gut-on-a-chips could form microvilli, whose shapes were different from villi; the flow field near the villi could not be investigated. Although ex vivo style devices have potential of the flow inside intestine thanks to the actual intestine, the flow field could not be measured due to the blockage of light by the intestinal wall. We found the lack of experimental technology for the observation of the flow field inside the small intestine, and therefore, development of a microfluidic device to measures the flow in the small intestinal tract is important. In this way, we explain the importance of our study including the research of live model. To improve the readability, we clearly revised the explanation in the introduction.
Although flow in the small intestine caused by peristalsis is inconstant, even a simple flow in the small intestinal tract (steady flow this time) has not been studied. As a first step, we developed a microfluidic device which can visualize the flow in the intestinal tissue, and apply a simple flow. This paper is the first report that a device that can examine the flow in the small intestinal tract and we plan to apply the inconstant flow caused by peristaltic movement in the future. We added the explanation that this study applied a constant flow as a simple model in section 3.3 and will apply an inconstant flow for the future work in conclusion.

Reviewer 2 Report
Dear authors,
This second version was much improved and I am pleased with the response to my comments. I found only a few minor spelling errors:
- line 56+58: 'enables' should be 'enable'
- line 61: 'the self-digestion' should be 'self-digestion'
- line 125: still mentions the word scraped
- line 259: the word 'order' seems to be misplaced or otherwise the sentence is incomplete
- line 264: does them mean the GB/particles? You might replace the word them by the word it refers to.
- line 309: 'an' should be placed between 'have' and 'important'
- line 309: 'the future works' should be 'future works'
Author Response
We really thank you for your valuable comments. Please see the attachment.

Reviewer 3 Report
It was confirmed that authors revised the manuscript along with the given comments.
Figure 10A is not unclear. It is better to provide the new pictures with high resolution.
Therefore, it is recommend to be accepted for publication after this revision.
Author Response

(The authors gave the same response as above.)

Round 3
Reviewer 1 Report
While somethings in this report are getting more accurate in the description of what was done, there are still critical flaws in some of the author’s assumptions. 1. This reviewer does not know the origin of the authors assumption that the gut wall would commence “self-digestion” immediately. While there are matrix metalloproteases in the lumen (bacterial source) and the tissue, they do not destroy the structure of the gut wall as long as the gut is maintained in a physiologically supportive medium. The gut wall is still better at surviving ex vivo longer than many other tissues in the body as long as it is kept at 4oC. It is not clear what the authors mean by “digestive enzymes”. Many enzymes in the gut lumen target nutritional sources and would not automatically start destroying whole cells. Lysosomal enzymes in the wall cells would not be released until cells started dying in large numbers and that does not start immediately. 2. Paraformaldehyde fixation would fail to lock glycoconjugates in place, so it is likely that significant mucus would be lost and that is a critical component of gut wall biomaterial characteristics. 3. “we believe that there is no expansion caused by the swelling of DI water”. The authors should take 2 sections of fixed intestinal tissue side x side and put one in isotonic saline and the other in DI water and then come back and say whether there is no expansion after fixation. I have done this experiment by accident and the expansion is shocking. 4. The authors seem to assume that the lengths of villi are the same across the segments of the small intestine. Not true. They are strikingly shorter in the ileum. Together with differences in Peyer’s patches that are smooth, you should predict that flow in isolated ileum should be significantly different than duodenum or jejunum.Author Response
We really appreciate your informative comments. Please see the attachment.
